# Formula Fertilization Promotes *Phoebe bournei* Robust Seedling Cultivation

**Zhi-Jian Yang [1,2], Xiao-Hui Wu [2], Steven C. Grossnickle [3,*], Lai-He Chen [2], Xin-Xiao Yu [1], Yousry A. El-Kassaby [4] and Jin-Ling Feng [2,*]**

[1] School of Soil and Water Conservation, Beijing Forestry University, Beijing 100083, China; yangzhijian@fafu.edu.cn (Z.-J.Y.); uxinxiao@bjfu.edu.cn (X.-X.Y.)

[2] College of Forestry, Fujian Agriculture and Forestry University, Fuzhou 350002, China; 3170422017@fafu.edu.cn (X.-H.W.); chenlaihe@fafu.edu.cn (L.-H.C.)

[3] NurserytoForest Solutions, North Saanich, BC V8L 5K7, Canada

[4] Department of Forest and Conservation Sciences, Faculty of Forestry, University of British Columbia, 2424 Main Mall, Vancouver, BC V6T 1Z4, Canada; y.el-kassaby@ubc.ca

[*] Correspondence: sgrossnickle@shaw.ca (S.C.G.); fengjinling@fafu.edu.cn (J.-L.F.)

**Abstract:** *Phoebe bournei* is a rare and endangered woody species and the success of its plantation development is dependent upon proper seedling cultivation. This study explored the regulation of N, P and K fertilizer and the interaction of these macronutrients on the growth of *Phoebe bournei* seedlings. To determine the optimum rate and ratio of N–P–K fertilizer in seedling cultivation, we used the unique "3414" incomplete orthogonal regression design to evaluate the effects of N–P–K fertilization on seedling morphological development. One-year-old *Phoebe bournei* bareroot seedlings were grown for one growing season under the defined fertilization regime, with their morphological development determined by measuring seedling attributes—root, stem, leaves and total biomass, root collar diameter and seedling height. These attributes were then combined to calculate the following indices: height-diameter ratio, shoot-root ratio and the seedling quality index (QI). Results showed that the N–P–K fertilizer had significant and beneficial effect on seedling cultivation. N effect was highest, followed by K and P. The three-way N×P×K interaction effect was strong, and the two-way interactions effect was highest for N×P, followed by P×K and N×K. At the "2" level of N (0.532 g·plant$^{-1}$), P (P$_2$O$_5$, 0.133 g·plant$^{-1}$), and K fertilizer (K$_2$O, 0.356 g·plant$^{-1}$), seedling growth and biomass accumulation were at their maximum. Unary, binary, and ternary quadratic fertilizer effect function equations of QI were established. Through comparative analysis, the ternary quadratic model was the optimal model and through a simulation–optimization, the optimal N–P–K fertilizer rates were 0.373~0.420 g·plant$^{-1}$ (N), 0.086~0.106 g·plant$^{-1}$ (P$_2$O$_5$), 0.243~0.280 g·plant$^{-1}$ (K$_2$O), with a N–P–K ratio of 1:0.20:0.43~1:0.65:0.75.

**Keywords:** *Phoebe bournei*; formula fertilization; "3414" fertilizer experiment; fertilization function model; seedling cultivation; morphological attributes

## 1. Introduction

*Phoebe bournei* (Hemsl.) Y.C. Yang (*Lauraceae*) is a subtropical evergreen broad-leaved tree sporadically distributed in China's southeast and southwest (Zhejiang, Fujian, Jiangxi, Guangdong, and Guangxi provinces) mountains, occupying an elevation band of 200–1000 m [1,2]. Due to its rare and endangered status, the species is under second-class protection in China [3]. *Phoebe bournei* wood has an exquisite texture and unique aroma, thus it is valued as precious material for high-end furniture, craft carving, and precision wood molding [2]. Because of the species high economic, social,

and ecological values, along with its unmanaged utilization, development of *Phoebe bournei* plantations have gained increased attention [4]. The species long growth cycle and low natural regeneration hampers its plantation production, leading to substantial deficit between market demand and wood supply [5]. As a result, there is a need to develop large *Phoebe bournei* plantation programs across China and the key to plantation establishment is the development of a robust forest regeneration system supported by scientifically-based seedling cultivation [6,7].

The success of forest regeneration programs depends on the development of nursery cultural protocols that produce quality seedlings. Seedling nutrient status has long been considered an important attribute in defining the morphological development of container-grown seedlings [8,9] and their quality at outplanting [10–12]. The importance of seedling morphological attributes has long been recognized [13], with Wakeley [14] as the first researcher to define morphological attributes that described seedling field performance. Morphological attributes are considered reliable measures of seedling quality because they are retained for extended timeframes after seedlings are outplanted [15]. Various reviews have defined the importance of morphological attributes [16,17], stating that seedlings should have a height within a defined range, a sturdy stem with at least a minimum diameter, a large fibrous root system and a well-balanced shoot-to-root system; with these attributes defining good seedling field performance [18,19]. The seedling quality index (QI), which combines morphological attributes along with biomass attributes, was designed as an integrated index of seedling quality [20]. The quality index proposed by Dickson et al. [20] was devised to forecast how well the combination of a number of shoot and root morphological attributes define seedling development and this index has been used to forecast subsequent field performance [16,21]. This, seedling quality index is an ideal way to measure the development of a proper fertilization regime for *Phoebe bournei* nursery-grown seedlings production.

In recent years, several *Phoebe bournei* seedlings fertilization studies were conducted producing varying results, mainly reflecting the various approaches used, such as different seedling age, experiment method, and other experimental conditions [22–24]. These inconsistent conclusions do not lend themselves to seedling production system generalizations and, more specifically, they were mainly focused on fertilizer application rate and did not deal with the application of nitrogen (N), phosphorus (P), and potassium (K) and their interaction. Therefore, we explored the effects of various N, P and K fertilizer application rates, and the interaction of these macronutrients on the morphological development of *Phoebe bournei* seedlings.

The "3414" experiment design, as a complete three-factor test, has the advantages of an optimal regression design with less treatments and a higher efficiency, thereby creating regression equations of unitary, binary and ternary secondary fertilizer effect [25]. It is widely used in field experiments to determine the fertilizer effect on plant performance [26–28]. Trials examining the simultaneous response to multi-nutrient fertilization treatments can produce results that are difficult to predict due to nutrient interactions. The "3414" fertilizer experimental design allows for the examination of multi-nutrient fertilization treatments, through the advantages of regression optimal design with less treatments and high efficiency. This approach can establish regression equations of fertilizer effects, it is more intuitive, allows for treatment comparability and is well suited for conducting multiple treatment field fertilization experiments [25–28]. The "3414" fertilizer experimental design was used to investigate N–P–K application rates and ratio effects on the growth of one-year-old *Phoebe bournei* seedlings. By fitting these equations of fertilizer effects, along with combining the theoretical optimization model and computer simulation optimization, we examined the N–P–K fertilization interaction effect on *Phoebe bournei* seedling development. This approach was intended to provide an optimization of nursery fertilization practices for *Phoebe bournei* seedlings. The main objective of the present study is to identify the fertilization treatment combination that optimizes the development of a robust *Phoebe bournei* seedlings growth and production. The seedling quality index (QI) was used as the main parameter for treatment comparison, thus the following section focuses on those attributes

associated with QI; namely, seedling growth (height and root collar diameter) and biomass attributes (root, stem and leaves).

## 2. Materials and Methods

### 2.1. Site Description

The experiment was conducted at the Fujian Agriculture and Forestry University, Fuzhou, Fujian Province, China (119°23′ E, 26°09′ N) field nursery in 2018. The nursery is equipped with a 2.7 m high shading shed providing a light intensity equivalent to 75% natural light. The environmental conditions during the trial (i.e., average monthly temperature and precipitation) are described in Table 1. The effective accumulated temperature ≥10 °C was 5880 °C, with a 326 frost-free-day period.

**Table 1.** The experiment environmental condition in 2018.

| Month | Average Temperature (°C) | Average Precipitation (mm) |
|-------|--------------------------|----------------------------|
| 3 | 16.7 | 98.1 |
| 4 | 20.3 | 71.5 |
| 5 | 25.5 | 156.2 |
| 6 | 26.4 | 214.3 |
| 7 | 29.1 | 173.3 |
| 8 | 28.3 | 301.5 |
| 9 | 27.0 | 124.3 |
| 10 | 21.5 | 61.2 |
| 11 | 15.2 | 130.5 |
| 12 | 10.6 | 25.1 |

### 2.2. Materials

We used one-year-old *Phoebe bournei* bareroot seedlings provided by the Fujian Academy of Forestry. In March 2018, well-grown seedlings with almost the same height (≈20.2 cm) and diameter (≈2.3 mm) with well-developed buds, were selected and planted (Figure 1). These seedlings had total N content of 1.219 g·Kg$^{-1}$, P content of 1.555 g·Kg$^{-1}$ and K content of 14.0222 g·Kg$^{-1}$ at the start of the experiment.

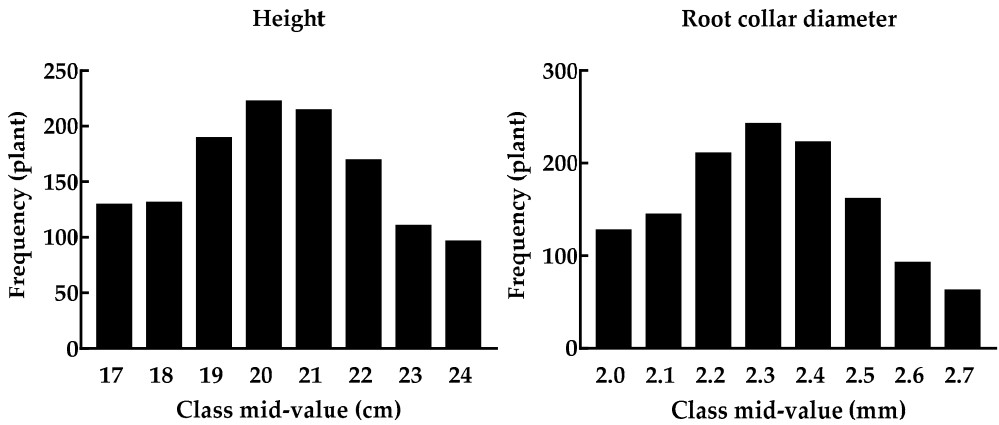

**Figure 1.** The height and root collar diameter distribution of *Phoebe bournei* seedlings as experiment materials.

The soil substrate was a mixture of red laterite soil, vermiculite and sand (6:2:2 by volume). The soil mixture was characterized as having 5.78 g·kg$^{-1}$ organic matter, pH 5.30, and total N–P–K content of 1.5 g·kg$^{-1}$, 0.035 g·kg$^{-1}$, and 33.91 g·kg$^{-1}$, respectively. Seedlings were grown in plastic

pots ($25 \times 25 \times 35$ cm). The test fertilizers were urea (N, 46.65%), superphosphate ($P_2O_5$, 12%) and potassium chloride ($KO_2$, 60%).

### 2.3. Experimental Design

The test followed the "3414" fertilizer experimental design (optimal regression design method) [25]. Tests were set as three factors with N, P and K. Every factor had four fertilization levels (0, 1, 2, 3): 0 as the control (no fertilizer); the common fertilizer rates as level 2; 0.5-times that of level 2 as level 1; 1.5-times that of level 2 as level 3 (see below). In total, there were 14 fertilizer treatments that can analyze the single factor, two-factor interaction and three-factor interaction effects of N, P, and K fertilization responses in relation to various seedling attributes, measured at the end of the trial. In the single-factor effect study, when the "2" level was fixed by P and K fertilizers, treatments T2–3, 6 and 11 were set as 0, 1, 2, 3 levels of N fertilizer application, respectively. When the "2" level was fixed by N and K fertilizers, treatments T4–7 were set to 0, 1, 2, and 3 levels of P fertilizer application, respectively. When the "2" level was fixed by N and P fertilizers, treatments T8–9, 6, and 10 were set to 0, 1, 2, and 3 levels of K fertilizer application, respectively. In a two-factor interaction study, when K fertilizer was fixed at a "2" level, the experiment for N–P interactions was set at eight treatments (T2, T3, T4, T5, T6, T7, T11 and T12). When the "2" level was fixed by P fertilizer, the experiment of N–K interaction was T2, T3, T6, T8, T9, T10, T11 and T13. When the N fertilizer was fixed at the "2" level, the experiment of P-K interaction was set at eight treatments (T4, T5, T6, T7, T8, T9, T10 and T14). The three-factor interaction experiment of N–P–K was set at T1–14 treatments. The specific amounts of fertilizer applied to each treatment are shown in Table 2, calculated in terms of N, $P_2O_5$, and $KO_2$. The basic fertilizer rates in the test were 0.532 g, 0.133 g and 0.356 g·plant$^{-1}$ for N, $P_2O_5$ and $KO_2$, respectively. Each treatment was repeated three times, with 30 plants in each treatment replication. The fertilizer replications were randomly arranged (complete randomized design) to minimize any environmental effects, as well as avoiding the clustering of a specific treatment.

**Table 2.** The 14 fertilizer rates used in the "3414" *Phoebe bournei* seedling growth fertilization experiment.

| Number | Fertilization Treatment | N (g·plant$^{-1}$) | $P_2O_5$ (g·plant$^{-1}$) | $KO_2$ (g·plant$^{-1}$) |
|---|---|---|---|---|
| T1 | $N_0P_0K_0$ | 0.000 | 0.000 | 0.000 |
| T2 | $N_0P_2K_2$ | 0.000 | 0.133 | 0.356 |
| T3 | $N_1P_2K_2$ | 0.266 | 0.133 | 0.356 |
| T4 | $N_2P_0K_2$ | 0.532 | 0.000 | 0.356 |
| T5 | $N_2P_1K_2$ | 0.532 | 0.067 | 0.356 |
| T6 | $N_2P_2K_2$ | 0.532 | 0.133 | 0.356 |
| T7 | $N_2P_3K_2$ | 0.532 | 0.200 | 0.356 |
| T8 | $N_2P_2K_0$ | 0.532 | 0.133 | 0.000 |
| T9 | $N_2P_2K_1$ | 0.532 | 0.133 | 0.178 |
| T10 | $N_2P_2K_3$ | 0.532 | 0.133 | 0.534 |
| T11 | $N_3P_2K_2$ | 0.798 | 0.133 | 0.356 |
| T12 | $N_1P_1K_2$ | 0.266 | 0.067 | 0.356 |
| T13 | $N_1P_2K_1$ | 0.266 | 0.133 | 0.178 |
| T14 | $N_2P_1K_1$ | 0.532 | 0.067 | 0.178 |

This field experiment lasted 10 months from March to December in 2018. P was used as the base fertilizer, while N and K were applied in different stages according to the seedling annual growth characteristics [29]. The applied fertilization regime varied over time (April: 25% N and 20% K; June: 35% N and 25% K; August: 25% N and 35% K; October: 15% N and 20% K) by liquid application of fertilizer (concentration 0.05%), any excess which leaked in the trays was promptly added back to the mixed matrix. Seedlings were watered on a varying schedule based on maintaining the desired soil water status of 75% field capacity (i.e., ~7 days during March to April, ~15 days during May to July, not watered in August, ~10 days during September to November, ~15 days in December).

*2.4. Measured Seedling Attributes*

In December 2018, after growth cessation, seedlings were measured for height and root collar diameter. The N, P, and K contents of seedlings are shown in Table 2. The N content was determined by the Kjeldahl method (ATN-300, Hongji, Shanghai, China) [30], the P content by the molybdenum-antimony colorimetric method (UV-2600A, Unicom, Shanghai, China) [31] and the K content by the atomic absorption spectrophotometer method (AA7002, Dongxi, Beijing, China) [32]. A random sample of three replications of three seedlings each per treatment (total of nine seedlings) were harvested to measure their biomasses. Seedlings were washed with water, divided into three parts (roots, stems and leaves), fixed at 105 °C for 15 min and then dried at 75 °C until reaching a stable weight. Samples' dry weights were measured with 0.001 g accuracy by electronic scales (AL204, Metiler-Toledo, Melbourne, Australia).

*2.5. Data Processing and Analysis*

2.5.1. Calculation of Comprehensive Index

Based on the seedling growth data (height, root collar diameter and roots, stems and leaves dry weights) the following indices were calculated:

- Seedling height–diameter ratio (HD) = seedling height (cm)/seedling root collar diameter (mm),
- Seedling shoot–root ratio (SR) = (stem biomass + leaves biomass) (g)/root biomass (g),
- Seedling quality index (QI) = TM/(HD + SR) [20,33].

2.5.2. Data Analysis

We used Excel 2010 to process the data. A factorial analysis of variance (ANOVA) was performed to evaluate the variance components and the Duncan's multiple range test for means multiple comparisons ($\alpha = 0.05$) by SAS 9.2 for the single factor, two-factor interaction and three-factor interaction effect of N, P and K fertilizer. SAS 9.2 statistical software was used to complete the full mathematical model information for the single factor, two-factor interaction and three-factor interaction effect of N, P and K fertilizer. During simulation optimization analysis and on the basis of the ternary quadratic regression model of QI value, an optimization analysis was carried out by taking seven code levels between N, $P_2O_5$ and $K_2O$ for a total of 488 possible combinations, obtaining 156 simulated experiments with a QI value greater than the mean (0.402), accounting for 31.97% of the total schemes and meeting the requirement of a large sample scheme by SAS 9.2 statistical software. The frequency analysis method was used to calculate the frequency of N, P and K application and the fertilization scheme of QI value > the average that can be conducive to the robust seedling cultivation was obtained. Then the 95% confidence interval was optimized for fertilization combinations [34]. Graphs were constructed with Origin 2018 (Systat Software, Inc., Washington, DC, USA) software.

## 3. Results

*3.1. Effects of Different Fertilization Treatments on Seedling Growth*

The different fertilizer application ratios significantly affected *Phoebe bournei* seedlings height, root collar diameter, and height–diameter ratio (HD), as shown in Figure 2. For height growth, T2, T3, T7 and T13 resulted in the greatest growth. For root collar diameter, T13, T3, T7, T6, T9, T10 and T14 resulted in the greatest growth. For HD ratio, there were T6, T14, T9, T1, T5 and T13 in the range 5–7.5. For seedling growth, T13 and T6 were highest with N, P and K content (N content: 13.34 g·kg$^{-1}$ and 14.02 g·kg$^{-1}$; P content: 3.39 g·kg$^{-1}$ and 4.00 g·kg$^{-1}$; K content: 44.55 g·kg$^{-1}$ and 37.48 g·kg$^{-1}$, respectively), as shown in Table 3. Different N concentrations produced different responses for the studied attributes (seedling height: $N_1 > N_0 > N_3 > N_2$; root collar diameter: $N_1 > N_2 > N_0 > N_3$; HD: $N_0 > N_1 > N_3 > N_2$), indicating that medium N fertilizer can reduce the growth and HD while

high-level reduced the root collar diameter. P treatments also produced different responses for the studied attributes (seedling height: $P_3 > P_0 > P_1 > P_2$; root collar diameter: $P_3 > P_2 > P_1 > P_0$; HD: $P_0 > P_3 > P_1 > P_2$), indicating that an increasing P application rate was associated with increase in root collar diameter and seedling height, while medium P fertilizer reduced HD. K treatments (seedling height: $K_3 > K_1 > K_0 > K_2$; root collar diameter $K_2 > K_1 > K_3 > K_0$; HD: $K_3 > K_0 > K_1 > K_2$) showed that medium K application rate can reduce seedling height growth, increase root collar diameter growth and thereby reduce HD. These results show that, for the most part, a moderate N, P and K ratio can improve *Phoebe bournei* seedlings shoot attributes. The greatest ranges of seedling height, root collar diameter and HD between the lowest and highest nutrient application rate were associated with N fertilizer application, followed by P and K, as shown in Table 4. In summary, N fertilizer had the greatest influence on seedlings growth followed by P and K fertilizer.

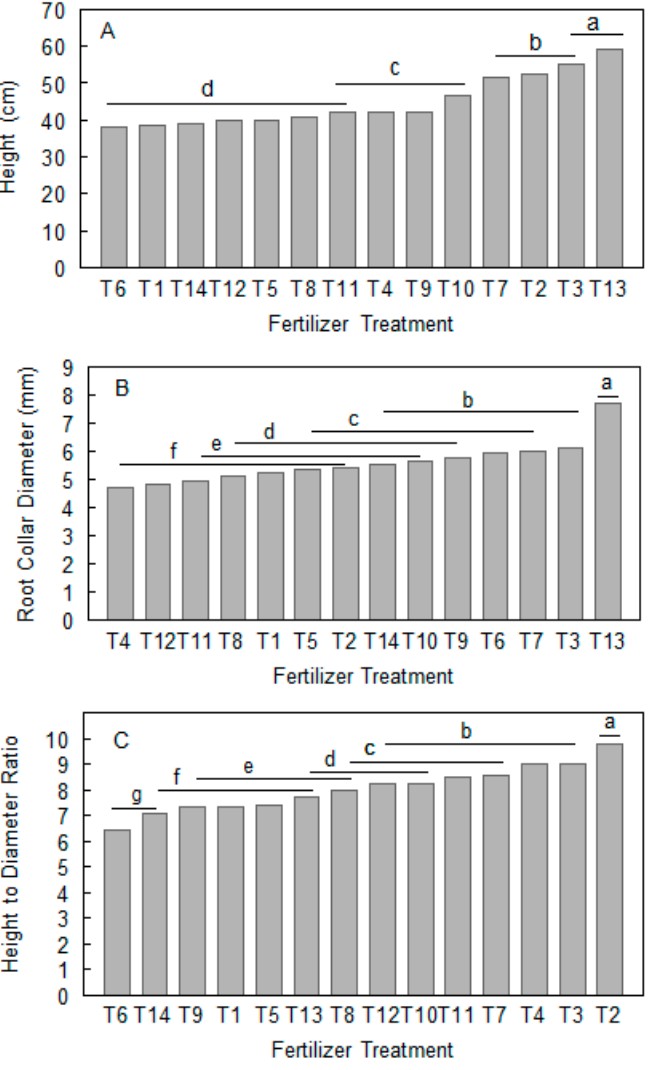

**Figure 2.** *Phoebe bournei* seedling growth attributes: (**A**) height (SE range 5.3 to 0.0); (**B**) root collar diameter (SE range 0.62 to 0.01); (**C**) height to diameter ratio (SE range 0.89 to 0.01). The means for different fertilization treatments—T1 to T14, as defined in Table 2—are listed in ascending order of treatment response. Treatment means sharing the same letter are not significantly different at ($p > 0.05$) as determined by the Duncan's multiple range test. Note: A height to diameter ratio in the range 5–7.5 provides seedlings with desirable shoot strength properties [17].

**Table 3.** *Phoebe bournei* seedling N, P, and K contents at the end of the trial for the 14 fertilization treatments. Details of the fertilization treatments are found in Table 2.

| Fertilization Treatment | N (g·kg$^{-1}$) | P (g·kg$^{-1}$) | K (g·kg$^{-1}$) |
|---|---|---|---|
| T1 | 10.88 | 3.16 | 27.11 |
| T2 | 14.60 | 4.32 | 48.38 |
| T3 | 13.84 | 3.74 | 39.52 |
| T4 | 12.62 | 2.56 | 23.15 |
| T5 | 15.84 | 3.97 | 53.81 |
| T6 | 11.34 | 3.39 | 44.55 |
| T7 | 12.49 | 3.33 | 25.58 |
| T8 | 13.77 | 2.81 | 23.24 |
| T9 | 16.33 | 4.05 | 36.73 |
| T10 | 13.55 | 3.15 | 50.71 |
| T11 | 17.48 | 4.68 | 40.70 |
| T12 | 13.43 | 3.67 | 38.75 |
| T13 | 14.02 | 4.00 | 37.48 |
| T14 | 15.25 | 2.98 | 31.85 |

**Table 4.** Range analysis of *Phoebe bournei* seedlings attributes measured at the end of the trial in response to the 14 fertilization treatments.

| Indicator [1] | N | P | K |
|---|---|---|---|
| Height (cm) | 17.12 | 13.55 | 8.58 |
| Root collar diameter (mm) | 1.16 | 1.31 | 0.8 |
| Rb (g) | 1.9 | 1.3 | 1.66 |
| Sb (g) | 1.44 | 1.07 | 1.26 |
| Lb (g) | 1.27 | 0.8 | 1.08 |
| Tm (g) | 4.55 | 3.17 | 3.97 |
| HD | 3.38 | 2.58 | 1.84 |
| Sr | 1.52 | 0.49 | 1.74 |
| QI | 0.66 | 0.45 | 0.57 |

[1] N: effect of N (0, 1, 2, 3) levels at P and K "2" level; P: effect of P (0, 1, 2, 3) levels at N and K "2" level; K: effect of K (0, 1, 2, 3) levels at N and P "2" level; HD: Height–diameter ratio; Sr: Shoot–root ratio; QI: seedling quality index.

*3.2. Effects of Different Fertilization Treatments on Seedling Biomass*

As with the seedling growth attributes, fertilizer application ratios significantly affected seedlings biomass attributes (root, stem, leaves and whole plant dry matter accumulation), as shown in Figure 3. T6, T4, T7, T13 and T12 produced the most seedling biomass and T6, T12, T4, T7, T9, T5, T1 and T14 had a SR ratio <2.5. For seedling biomass, they were highest with N, P and K contents of 13.34 g·kg$^{-1}$, 3.39 g·kg$^{-1}$ and 44.55 g·kg$^{-1}$, respectively, as shown in Table 3. Generally, different N, P and K concentrations produced different responses for the biomass attributes. In summary, the moderate increase in N, P and K application rates was associated with an increase in the dry matter accumulation in root, stem, leaves and the whole plant, and an excessive increase in fertilization produced a negative response. The dry matter accumulation of root, stem, leaves and whole plant, and SR ratios in T12–14 were higher than T1, showing that a moderate combination of N, P and K benefited seedlings dry matter accumulation, and more in the above-ground growth than the underground part. The range between the lowest and highest nutrient application rate of various biomass accumulation indices was observed for N, followed by K and P, while for the SR ratio it was K, followed by N and P, as shown in Table 4. This indicated that N fertilization had the greatest impact on biomass accumulation and K represents the most important nutrient that controlled the above- to below-ground growth relationship.

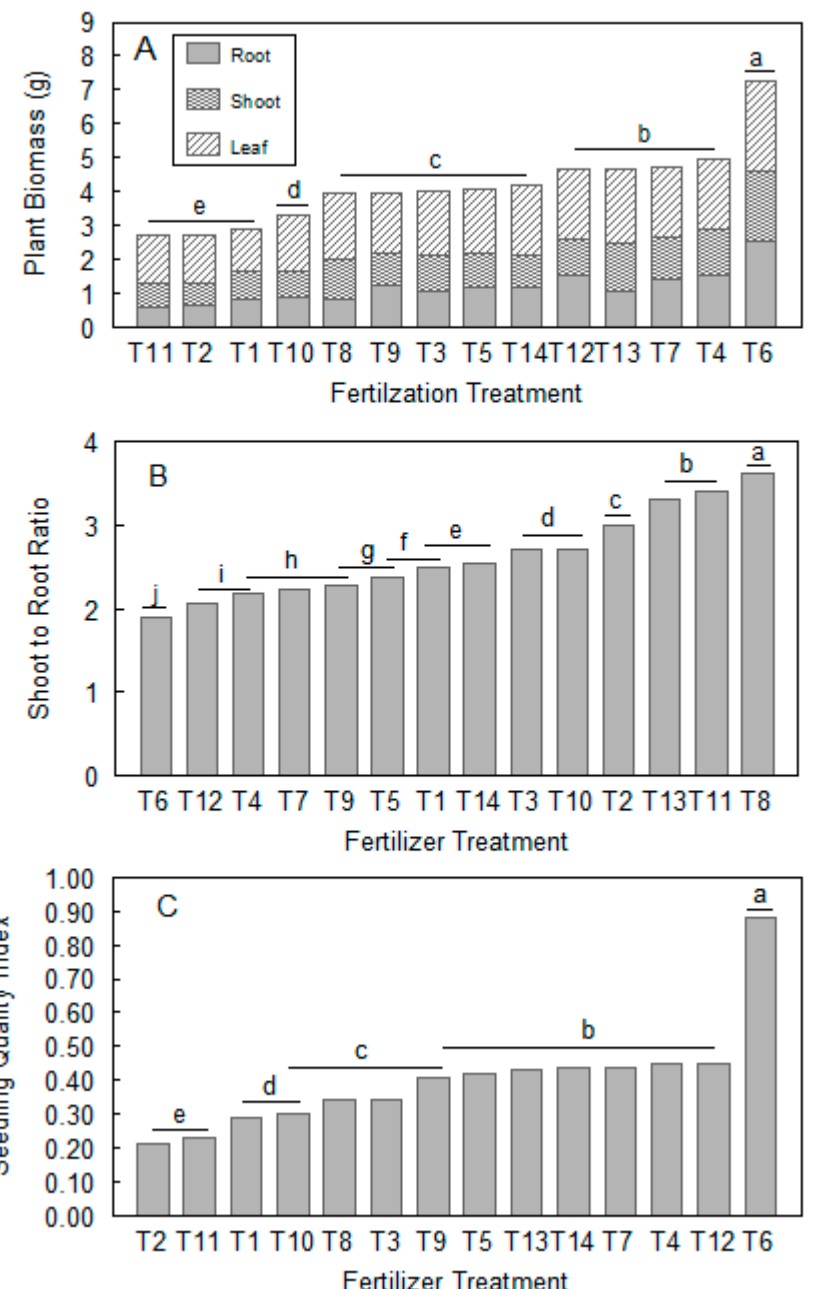

**Figure 3.** *Phoebe bournei* seedling growth attributes: (**A**) plant biomass (SE range 0.41 to 0.07); (**B**) shoot to root ratio (SE range 0.18 to 0.01); (**C**) seedling quality index (SE range 0.04 to 0.01). The means for different fertilization treatments—T1 to T14 as defined in Table 2—are listed in ascending order of treatment response. Treatment means sharing the same letter are not significantly different at ($p > 0.05$), as determined by the Duncan's multiple range test. Note: Shoot-to-root ratios <2.5 are considered desirable for good seedling field performance [18,19].

### 3.3. Effects of Different Fertilization Treatments on Seedling Quality Index (QI)

The different fertilization treatments significantly affected the seedling quality index (QI), as shown in Figure 3. Treatments T6, T12, T4, T7, T14, T13 and T5 gave the highest SQI. N, P and K fertilizer applications at low and medium levels improved QI and any increase was associated with a negative effect, indicating that the moderate application of N, P and K fertilizer could increase the QI. The range between the lowest and highest nutrient application rate found that N produced the greatest effect, followed by K and, finally, P on QI value, as shown in Table 4, indicating the importance of N fertilizer

relative to K and P. All first, second and third order interactions of N, P and K fertilizer produced highly significant QI values ($p < 0.01$), as shown in Table 5. Therefore, unitary, binary and ternary quadratic regression analyses and the modeling of the QI value were performed for N, P and K fertilization.

**Table 5.** ANOVA of the "3414" *Phoebe bournei* seedling growth response in the fertilization experiment, as defined by a measure of the seedling quality index (QI).

| Source of Variation | df | SS | MS | *F* Value | *p*-Value |
|---|---|---|---|---|---|
| N | 3 | 0.88 | 0.29 | 386.88 | <0.001 |
| P | 3 | 0.44 | 0.15 | 162.70 | <0.001 |
| K | 3 | 0.64 | 0.21 | 289.15 | <0.001 |
| N×P | 7 | 0.89 | 0.13 | 176.96 | <0.001 |
| N×K | 7 | 0.92 | 0.13 | 196.83 | <0.001 |
| P×K | 7 | 0.66 | 0.09 | 157.31 | <0.001 |
| N×P×K | 13 | 0.44 | 0.05 | 122.15 | <0.001 |

### 3.4. Correlations among N–P–K Application Rates with Seedling Attributes and QI Index

All fertilizer applications produced positive correlations with QI value, with decreasing correlation coefficients in the following order: N > K > P, as shown in Table 6. N fertilizer application rate was negatively correlated with SR, height, root collar diameter and HD (significantly with height), and positively correlated with root, stem, leaves and total biomass (significantly with leaves biomass). The P fertilizer application rate was negatively correlated with root biomass and positively correlated with other indicators (significantly with height and root collar diameter). The K fertilizer application rate was negatively correlated with SR and root collar diameter (significantly with SR), and positively correlated with other indicators.

**Table 6.** Correlation analysis of fertilizer application rate and seedling quality index (QI) value in relation to other plant attributes measured at the end of the trial for *Phoebe bournei* seedling growth response to the 14 fertilization treatments.

| Index | N | $P_2O_5$ | $KO_2$ |
|---|---|---|---|
| Root biomass | 0.20 | −0.02 | 0.22 |
| Stem biomass | 0.19 | 0.09 | 0.01 |
| Leaves biomass | 0.30 * | 0.17 | 0.10 |
| Total biomass | 0.24 | 0.07 | 0.13 |
| Shoot–root ratio | −0.02 | 0.29 | −0.32 * |
| Height | −0.31 * | 0.50 ** | 0.20 |
| Root collar diameter | −0.15 | 0.44 ** | −0.06 |
| Height–diameter ratio | −0.23 | 0.13 | 0.35 * |
| QI [1] | 0.23 | 0.04 | 0.09 |

[1] QI, seedling quality index; * $p < 0.05$; ** $p < 0.01$.

### 3.5. Interaction Analysis of N–P–K Fertilizer Applications

The first order interactions between N, P and K application rates (i.e., N×P, N×K and P×K) produced significant influence on seedling QI value, as shown in Table 5, thus they were further analyzed using binary quadratic equation fitting equations, as shown in Table 7, and the 3D surface diagrams facilitated their interpretation, as shown in Figure 4. The effect of N–P–K fertilizer applications on seedling QI value was parabolic with increasing trend that followed with a decrease, consistent with the law of diminishing returns. For P×K interaction, the increment effect of P and K first increased and then decreased with the increase in the application level; however, the influence of K on seedling QI value was large and the curve changed sharply with little effect associated with change in P rate, as illustrated by the relatively flat curve, indicating that P can promote the effect of K application, as shown in Figure 4A. For N×K interaction, the QI value increased with the increase in and K application level,

indicating a mutual promotion, as shown in Figure 4B. The QI value reached the maximum range at medium N and K levels and decreased with the increase in fertilizer application, as shown in Figure 4B. For N×P interaction, the QI value increased with the increase in N and P application and reached the maximum range at the medium level of N and P; however, the curve of N changed dramatically, while remaining relatively flat for P, indicating that P can promote a positive effect on N application, as shown in Figure 4C.

**Table 7.** N–P–K fertilizer response equations and the recommended fertilizer application rates for optimum *Phoebe bournei* seedling growth.

| Model | Nutrient | Fertilizer Response Equation | $p$ | $R^2$ | Maximum Rate (g·Plant$^{-1}$) |
|---|---|---|---|---|---|
| | N | $Y = 0.134 + 2.409N - 2.742N^2$ | 0.023 | 0.686 | 0.439 |
| Unitary | $P_2O_5$ | $Y = 0.376 + 5.309P - 23.314P^2$ | 0.147 | 0.448 | 0.114 |
| | $K_2O$ | $Y = 0.270 + 2.931K - 5.128K^2$ | 0.037 | 0.641 | 0.286 |
| | N | $Y = -0.202 + 2.359N + 4.402P -$ | 0.030 | 0.570 | 0.442 |
| | $P_2O_5$ | $2.297N^2 - 13.089P^2 - 2.648NP$ | | | 0.123 |
| Binary | N | $Y = -0.421 + 2.555N + 2.938K -$ | 0.025 | 0.582 | 0.458 |
| | $K_2O$ | $2.269N^2 - 3.724K^2 - 1.611NK$ | | | 0.295 |
| | $P_2O_5$ | $Y = -0.296 + 6.376P + 3.385K -$ | 0.110 | 0.442 | 0.136 |
| | $K_2O$ | $14.800P^2 - 4.025K^2 - 8.413PK$ | | | 0.278 |
| | N | $Y = 0.284 + 0.936N + 0.076P + 0.594K$ | | | 0.361 |
| Ternary | $P_2O_5$ | $- 1.776N^2 - 4.786P^2 - 2.623K^2 +$ | 0.016 | 0.529 | 0.061 |
| | $K_2O$ | $3.348NP + 0.575NK + 1.442PK$ | | | 0.246 |

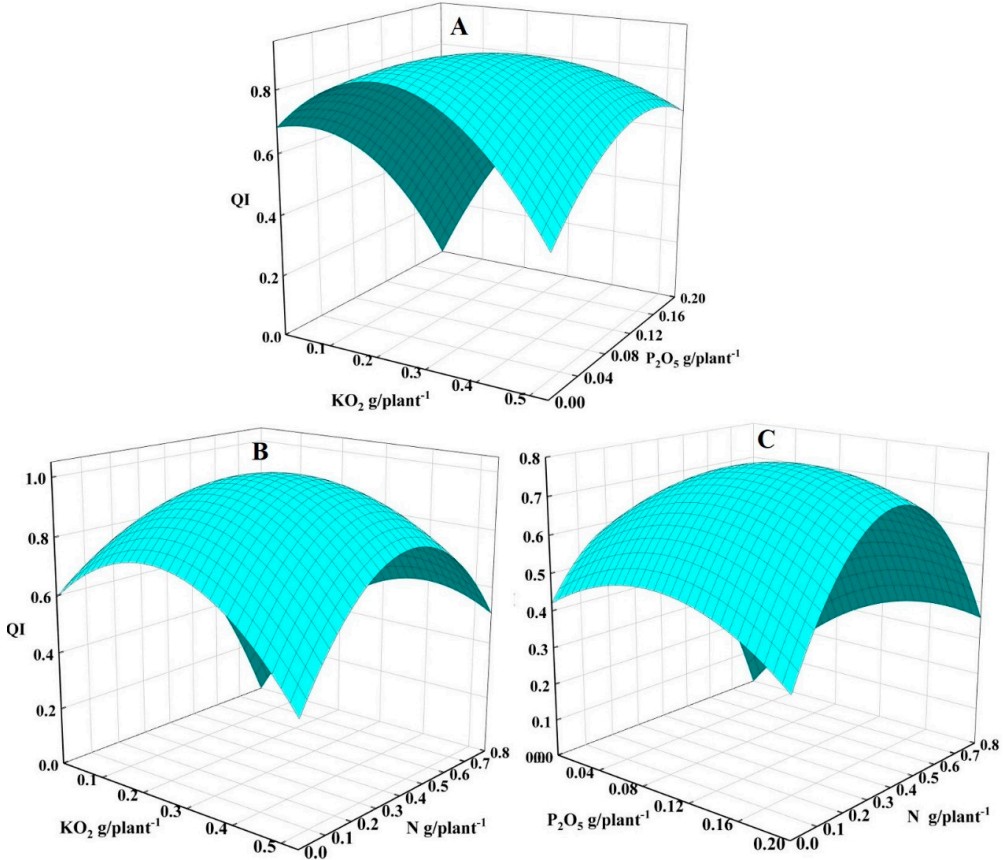

**Figure 4.** The seedling quality index (QI) for *Phoebe bournei* seedlings in response to the N, P, and K fertilization interactions of: (**A**) N×P; (**B**) N×K; (**C**) P×K.

## 3.6. Fertilizer Function Model Fitting and Fertilizer Rate Analysis

The QI value results of the unitary, binary and ternary quadratic equations of N–P–K produced significant ($p < 0.05$) N, K, N–P, N–K and N–P–K models and non-significant ($p > 0.05$) P and P–K models, confirming lack of fit for P and P–K models, as shown in Table 7. Thus, it is estimated that the highest N, P and K application rates for *Phoebe bournei* seedlings were 0.439 g·plant$^{-1}$ (N), 0.123 g·plant$^{-1}$ (P$_2$O$_5$) and 0.295 g·plant$^{-1}$ (K$_2$O), respectively by all models.

The ternary quadratic regression model is dimensionless, and the absolute value of partial regression coefficients of the first degree of equation can directly reflect the relationship between N–P–K (N > K > P), and their interactions, respectively (NP > PK > NK). At the same time, the partial regression coefficients were positive, indicating that N–P–K have positive effects on robust seedling cultivation. It is estimated that the maximum fertilizer application rate for *Phoebe bournei* seedlings was 0.361 g·plant$^{-1}$ (N), 0.061 g·plant-1 (P$_2$O$_5$) and 0.246 g·plant$^{-1}$ (K$_2$O) by the ternary model, as shown in Table 7. The simulation optimization analysis of the ternary quadratic regression model of QI value, calculated 156 simulated experiments describing the frequency of the N–P–K application rate. The management plan (mean value of QI value > 0.402) was obtained by the frequency analysis method, as shown in Table 8. When the QI value was greater than the mean value (0.402), the corresponding optimal N–P–K fertilizer application rate was as follows: 0.373~0.420 g·plant$^{-1}$ (N), 0.086~0.106 g·plant$^{-1}$ (P$_2$O$_5$) and 0.243~0.280 g·plant$^{-1}$ (K$_2$O), meeting the optimal combination of 95% confidence interval. The ratio of N–P–K fertilizer was approximately between 1:0.20:0.43 and 1:0.65:0.75.

**Table 8.** Frequency distribution and fertilization scheme of N–P–K encoding values of the seedling quality index (QI) measured at the end of the trial for defining *Phoebe bournei* seedling growth response.

| Item | N | P$_2$O$_5$ | K$_2$O |
|---|---|---|---|
| Coding weighted average | 0.396 | 0.096 | 0.262 |
| Standard error | 0.148 | 0.064 | 0.116 |
| 95% confidence interval | 0.373~0.420 | 0.086~0.106 | 0.243~0.280 |
| Optimal fertilizer (g·plant$^{-1}$) | 0.373~0.420 | 0.086~0.106 | 0.243~0.280 |

## 4. Discussions

### 4.1. Effects of N–P–K Fertilizer on Phoebe bournei Seedlings Growth

The application of fertilizer is one of the main nursery cultural practices used to grow containerized seedlings [8,9]. In addition, seedling nutrition has long been considered an important attribute when viewed through the concept that fertilization practices can modify seedling morphological development in ways that are potentially beneficial for seedling field performance [10–12]. This is why nutrition is considered an important attribute in recent seedling quality discussions [35–37]. This trial examined a range of fertilization regimes for *Phoebe bournei* to determine which regime produced seedlings with desirable morphological attributes. The appropriate combined fertilization of N, P and K nutrition can enhance plant growth and biomass accumulation [38]. In this trial, a moderate N, P and K ratio improved *Phoebe bournei* seedlings shoot attributes of seedling height, root collar diameter, and HD. In addition, a moderate N, P and K application rate increased seedling dry matter accumulation. The availability of nutrient ions during the production of container-grown seedlings affects their growth [9,39,40]. Tree species growth response to nutrient availability spans from limiting growth due to nutrient deficiency, to optimum growth (i.e., optimum nutrient range), towards luxury consumption (i.e., nutrients are available but do not result in greater growth) and reduced growth (i.e., toxic range) if nutrient concentrations reach extremely high levels [9,40–42]. The range of fertilization treatments assessed in this study enabled us to define the optimum nutrient range to maximize *Phoebe bournei* seedling growth.

Correct proportionality between shoot and root systems is a desirable plant attribute and the seedling quality index (QI) is one way to assess seedling morphological balance [33]. In the present trial, a moderate application of N, P and K fertilizers could increase the QI. This indicates that the moderate fertilization treatment of T6 (N content: 13.34 g·kg$^{-1}$; P content: 3.39 g·kg$^{-1}$; K content: 44.55 g·kg$^{-1}$) was the optimum nutrient range [41] for growing *Phoebe bournei* seedlings. It is generally accepted that the optimum nutrient range for N, P and K is associated with enhanced shoot growth over root growth [43]. Thus, increasing seedling nutrient content beyond the optimum range may increase the shoot-to-root ratio or lower the QI value. The N effect was highest, followed by K and P fertilizer (N > K > P) with a positive correlation with the QI value. However, individual applications had different effects on growth and sometimes even inhibited growth [44]. This shows that excessive fertilization reduced *Phoebe bournei* seedling growth if nutrient concentrations reached a high enough level to be within a toxic range [41].

### 4.2. Effects of N, P, K Fertilizer on Phoebe bournei Seedlings Growth

The single-nutrient effect, under the second level of two other fertilizers not only showed a response change in fertilizer application rates, but also reflected the change in nutrient proportion and their interactions. With an increase in N applied, height, root collar diameter and dry matter accumulation in root, stem, leaves and whole plant first increased and then increased correspondingly, indicating that *Phoebe bournei* seedling growth was mainly caused by the rate of N applied. With an increase in P applied, root collar diameter increased and height at first decreased and then increased correspondingly, but the dry matter accumulation in root, stem, leaves and whole plant first increased, decreased and then increased. This indicates that the P fertilization rate affected root collar diameter and height growth, while dry matter accumulation was affected by the N–P–K ratio and its interaction. With an increase in K applied, root collar diameter and total dry matter accumulation first decreased and then increased correspondingly, though height first increased, decreased and then increased. The K application rate affects root collar diameter growth and dry matter accumulation, while height growth was also affected by the N–P–K ratio and its interaction.

The N fertilizer application rates positively correlated with root, stem, leaves and total biomass and QI, among which leaves biomass was significant, confirming the role of N in promoting biomass accumulation, especially that of leaves [36,43]. Additionally, N fertilizer rates were negatively correlated with the seedling shoot–root ratio, height, root collar diameter, and height–diameter ratio, among which height was significantly correlated, indicating that N fertilizer inhibited diameter and height growth (especially height growth), and changed the biomass allocation among *Phoebe bournei* seedling parts [45,46]. Seedling shoot balance [17] and shoot-to-root balance [18,19] are critical plant attributes to consider to ensure good field performance after outplanting. Thus, one needs to apply a medium N fertilization rate to ensure a desirable *Phoebe bournei* seedling morphological balance. Tree species demand for N is closely related to their amount of growth [47]. Recent studies have shown N increased the photosynthetic carbon fixation capacity and promoted the accumulation of seedling biomass (root, stem, leaves and total biomass) [48,49]. For *Phoebe bournei* seedlings, the application of N changed the biomass allocation among seedling parts, which appeared to be beneficial to leaf growth, while relatively limiting to stem and root growth. This same phenomenon is reported in other trials [26,44,46].

P fertilization rates were positively correlated with height and root collar diameter, indicating that P promoted seedling growth, thus it is beneficial to aboveground biomass accumulation. Other nursery fertilization studies have found that optimum P levels promote seedling shoot [43] and total plant biomass accumulation [50–52]. Studies have also found that P fertilization can change biomass allocation among seedling parts, thereby limiting the accumulation of root biomass [53–55], results similar to those reported on *Phoebe bournei* seedlings [22]. K fertilizer promoted root collar diameter growth and a moderate increase in K application rate increased the dry matter accumulation in root, stem, leaves and the whole plant of *Phoebe bournei* seedlings, confirming K fertilizer can improve the photosynthetic capacity of leaves [42,47]. K fertilizer can reduce seedling height growth, and is significantly positively

correlated with height–diameter ratio, and significantly negatively correlated with the shoot–root ratio, confirming that K application promoted the stem growth of seedling and the transport of photosynthetic products underground by increasing the accumulation of root biomass [6,56]. K fertilizer increased the QI, showing that K application was beneficial to the growth of *Phoebe bournei* seedlings.

### 4.3. Interaction of N–P–K Fertilizer on Phoebe bournei Seedlings Growth

Notwithstanding, nutrient elements have major effects on seedling growth and their interaction effects are also extremely important to plant development. Through exploring the trajectory and magnitude of the interaction among plant nutrients and finding the appropriate interaction ratio, the fertilizer effect can be effectively maximized [57]. In the P×K interaction, as shown in Figure 4A, the highest point was 0.136 g·plant$^{-1}$ ($P_2O_5$) and 0.278 g·plant$^{-1}$ ($K_2O$); in the N×K interaction, as shown in Figure 4B, the highest point was 0.458 g·plant$^{-1}$ (N) and 0.295 g·plant$^{-1}$ ($K_2O$); in the N×P interaction, as shown in Figure 4C, the highest point was 0.442 g·plant$^{-1}$ (N) and 0.123 g·plant$^{-1}$ ($P_2O_5$), which was close to T6 (N, 0.532 g·plant$^{-1}$; $P_2O_5$, 0.133 g·plant$^{-1}$; $K_2O$, 0.356 g·plant$^{-1}$; N:$P_2O_5$:$K_2O$ = 1:0.250:0.669), which was the most favorable treatment for seedling morphological development. These results are consistent with findings reported for *Phoebe chekiangensis* and *Phoebe bournei* seedlings [24,58]. The range analysis also showed that, in N, P and K fertilizer combined applications, the greatest effects were on the growth, biomass accumulation and QI value, and highlighted the important role of N on growth and robust seedling cultivation. The N–P–K fertilizers interaction analysis showed that the effects of P and K fertilizer were promoted by the P fertilizer, and that N and K fertilizers promote each other, which had strong interaction effects on N×K, N×P and P×K fertilizers with N×P > P×K > N×K.

### 4.4. Optimum Application Rate of N–P–K Fertilizer for Seedlings Production

Measures of shoot-to-root (SR) balance define seedling drought avoidance potential [16]. Seedlings need to have root systems in a proper proportion to the shoot system. Typically, a SR ratio <2.5 results in the highest survival and subsequent seedling growth when outplanted [18,19], and there were T6, T12, T4, T7, T9, T5, T1 and T14. For the HD ratio, bigger is not better. Based on the cantilever beam concept (as a seedling extends its length, it must expand in DIA to maintain the same relative strength properties), thus having the HD in the range 5–7.5 is desirable [16,17]. Seedlings with a HD in this desirable range were T6, T14, T9, T1, T5 and T13. For height growth, T2, T3, T7 and T13 resulted in the greatest HT growth. For root collar diameter, T13, T3, T7, T6, T9, T10 and T14 resulted in the greatest growth. T6, T4, T7, T13 and T12 produced the most seedling biomass. T6, T12, T4, T7, T14, T13 and T5 gave the highest SQI. Summarizing the above results, T6 was considered the best fertilization treatment for growing *Phoebe bournei* seedlings. However, the fertilization effect of *Phoebe bournei* seedlings was not only affected by fertilization rate, but also by fertilization ratio and nutrient interactions. Only through the fertilizer function fitting analysis can we simultaneously analyze the amount of fertilizer applied, the proportion of fertilizer applied, nutrient interactions and, finally, determine the N, P and K fertilizer applied and their proportion suitable for the optimum growth of *Phoebe bournei* seedlings.

The fertilizer function model is the main method to determine the optimal application rates, and the selection of a preferred fertilizer model [59]. The unique "3414" experimental design not only has the advantage of using less treatments and maximizing the efficiency of the design's optimal regression, but it also allowed for meeting the necessary requirements of fertilizer test and decision. This design is widely used in determining the fertilizer effect of grain, vegetables and other crops with remarkable success [25,26,28]. In the present study, the fertilizer application rates calculated by unitary, binary and ternary models were within the present experimental range. However, in practice, the fertilizer effect function should give priority to the ternary quadratic equation [60], in which the QI value cannot be directly determined as the increase or decrease in the function under the influence of N, P and K fertilization factors [61]. Therefore, the method of frequency analysis was adopted to conduct simulation optimization analysis for this ternary quadratic equation model. This allowed us to determine that the

optimal fertilizer application rate was: 0.373~0.420 g·plant$^{-1}$ (N), 0.086~0.106 g·plant$^{-1}$ (P$_2$O$_5$) and 0.243~0.280 g·plant$^{-1}$ (K$_2$O) with an N:P:K ratio between approximately 1:0.20:0.43 and 1:0.65:0.75; results supported by the findings of Wang Y et al. [23].

## 5. Conclusions

This trial applied a unique "3414" experimental design to determine the optimal fertilization application rates of N, P and K for the nursery cultivation of *Phoebe bournei* seedlings. The QI was used as the main seedling quality attribute for treatments comparison because it integrated both seedling growth (height and root collar diameter) and biomass attributes (root, stem and leaves) to define performance in relation to the fourteen fertilization treatments assessed in this trial. The QI was used for fertilizer model fitting to determine the optimal N–P–K fertilizer rates for seedling production. This experimental approach found that the N–P–K fertilizer had a significant and beneficial effect on seedling cultivation. The N effect was highest, followed by K and P. The three-way N×P×K interaction effect was strong, and the two-way interactions effect was the highest for N×P, followed by P×K and N×K. When any two fertilizers were fixed at the "2" level, the effect of the third varied. At the "2" level of N (0.532 g·plant$^{-1}$), P (P$_2$O$_5$, 0.133 g·plant$^{-1}$) and K fertilizer (K$_2$O, 0.356 g·plant$^{-1}$), seedling growth and biomass accumulation were at their maximum. Unary, binary, and ternary quadratic fertilizer effect function equations of QI were established. Through comparative analysis, the ternary quadratic model was the optimal model and through a simulation–optimization, the QI values reached their optimum values of seedling development when the N–P–K fertilizer rates were 0.373~0.420 g·plant$^{-1}$ (N), 0.086~0.106 g·plant$^{-1}$ (P$_2$O$_5$) and 0.243~0.280 g·plant$^{-1}$ (K$_2$O), with an N–P–K ratio of between approximately 1:0.20:0.43 and 1:0.65:0.75. These results are of practical significance for optimizing seedling fertilization technology and the development of a robust *Phoebe bournei* seedling nursery production protocol.

**Author Contributions:** Conceptualization, S.C.G., X.-X.Y., Y.A.E.-K. and J.-L.F.; Data curation, Z.-J.Y., X.-H.W. and L.-H.C.; Formal analysis, Z.-J.Y.; Investigation, X.-H.W. and X.-X.Y.; Methodology, Z.-J.Y. and X.-H.W.; Project administration, J.-L.F.; Resources, J.-L.F.; Supervision, X.-X.Y. and J.-L.F.; Writing—original draft, Z.-J.Y. and X.-H.W.; Writing—review and editing, S.C.G. and Y.A.E.-K. All authors have read and agreed to the published version of the manuscript.

**Funding:** This study was funded by the Fujian Soil and Water Conservation Research Project (KH180280A) and Fujian Forest Seedling Science and Technology Research Project (KLh16H04A).

**Acknowledgments:** We are grateful to the Fujian Academy of Forestry for providing the *Phoebe bournei* seedlings.

**Conflicts of Interest:** The authors declare no conflict of interest.

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
