# Peer review of "Formula Fertilization Promotes Phoebe bournei Robust Seedling Cultivation"

_forests, doi:10.3390/f11070781_

Round 1

Reviewer 1 Report

Comments FOR-831137v2

General:

This manuscript reports on an interesting fertilizer design and analysis to determine the optimal fertilizer rate for a tree seedling; however, I have some serious concerns about how the experiment was carried out. The authors also have inadequate descriptions or justifications of the design, choice of materials and methods, and analyses. The results of the study are difficult to interpret and show trends or a lack thereof that are contrary to expectations, This makes it impossible to properly interpret the results or evaluate the potential flaws of the study. I have highlighted some of the major issues below.

Also, the manuscript contains simple writing errors throughout. It needs to be carefully proofread after it is revised.

Abstract:

The abstract does not specify the form or fertilizer applied to the plants or the length of time the plants were grown in the nursery. Both of these are important for interpreting the recommended fertilizer rates included at the end of the abstract.

Introduction

  • Lines 63-65. The seedling quality index (QI) needs to be more clearly described. Providing a couple of citations is not sufficient.
  • Lines 73-74. This sentence does not clearly distinguish what was done from what the authors suggest needs to be done. It’s also not clear what the authors are recommending be done. 
  • Lines 75-93. This paragraph needs improvement to properly describe the 3414 experimental design and to justify its preference to related designs. There are no related designs even mentioned, so it is unknown what this design is being compared to.

Materials and Methods

  • Site description. It would be helpful to include figures showing monthly averages or cumulative totals for temperature, precipitation, and solar radiation.
    • Sunshine hours is a climatological measurement, but it is typically used as an anthropocentric measure of how “sunny” vs “cloudy” a location is. Insolation is a more relevant measure for plants.
  • Materials. Again, it’s good the authors include average size for the seedlings, but a histogram or related figure showing the size distribution would be helpful as a supplement.
    • Lines 108-109. Do not put the actual nutrient concentrations in parentheses. The units should be g Kg-1 for plant nutrient concentration, not mg Kg-1. And the K concentration appears to be an order of magnitude too low, based on results in Table 2.
    • The red soil and sand need more explanation as to their mineralogy or taxonomy. The texture of the red soil would also be helpful. 
    • This is also a low pH for a potting mixture. Does this species tend to grow in acid soils, and this “red soil” in particular? Was the soil from a commercial source? Was it sterilized prior to use?
    • The size of the pots (25 x 25 x 35 cm) is extremely large for seedlings of this size (20 cm in height and 2.3 mm in root collar diameter). The 
  • Experimental design. As mentioned for the Introduction, the 3414 design rationale must be adequately described here. It seems to be used a lot in Chinese fertilization studies, but most of the articles are not written in English, and the few I found in English do not describe the rationale of this design or how it is analyzed.
    • The authors do not provide a concise description of the timing or length of the experiment. They mention May 2018 as the planting time (Line 106) but then mention April, June, August and October as fertilization dates (Lines 132-133). This info needs to be put together in a single paragraph, with specific dates and a total time period included.
    • The watering regime is also not described adequately. How did the authors estimate field capacity? How often and how much water was added to the pots during irrigation events?
  • Data analysis. The 3414 design is given a reasonable explanation in terms of the purpose for analysis as a factorial ANOVA. It is not clear why SAS and SPSS are used separately for the factorial analysis. Either should be able to accomplish this. It’s also not clear from the description how the analysis in SPSS is different from what was done in SAS.
  • In addition, the simulation and optimization procedures are not adequately described. The details need to be supported by an overall explanation of the procedure. This is a place where citations to prior work would be helpful to justify and support this analysis.

Results

    • The seedlings appear to have doubled in height and root collar diameter over the experimental period. This makes it critical that the authors specify how long the seedlings were grown in this experiment, given that they were 1 year old at the start of the experiment. They seem to have been relatively small after one year of propagation, only 20 cm in height and 2.3 mm in diameter. For a sub-tropical tree, this is very slow growth.
  • Effects of different fertilizer treatments. The results of single-nutrient effects are unexpected and in some cases seem nonsensical. Effects on height and biomass, in particular, show odd rankings and a lack of expected trends.
    • For example, for biomass, T6, which represents the recommended levels of N, P, and K fertilizer, is way larger than all other treatments. It ranks lowest in height, however, and it does not have the largest root collar diameter, even though it has by far the largest root biomass. The single-nutrient trends for N and P also do not follow any understandable ranking within the biomass data, so it’s not simply a nutrient interaction effect at work here.
    • The seedling quality results and the simulation-optimization results are highly influenced by the much larger root biomass of T6. The QI of this treatment is twice that of the other treatments, so for the optimization graphs (Fig. 3), this mid-point of fertilization is clearly going to show an optimum. Again, this might be expected, but the dramatic difference is highly questionable.
  • Correlations among application rates and QI. The results here illustrate the questionable lack of trends or counter-intuitive trends in the data for fertilization rates. The strong correlation of QI with biomass components and ratios is not informative. These components are part of the QI equation, so they would be expected to correlate strongly.
  • The fertilizer function fitting is not meaningful unless these other issues are addressed. The entire analysis is overly influenced by the biomass and QI results for T6.

Discussion

  • The discussion does little to address these concerns. 
  • Section 4.1 is mostly background information, some of which should have gone into the Introduction. 
  • Sections 4.2-4.4 re-report the results with insufficient interpretation of the trends or lack thereof. The recommendations are based on results that cannot be trusted or properly interpreted as reported.

Reviewer 2 Report

This work is aimed at  identification of the fertilisation treatment combination that optimize the development of Phoebe bournei seelings growth and production. The development of plantations of Phoebe bournei attracts much attention because this species has a high economical, cosial and ecological values but its utilization is poorly managed.

The authors obtained new, interesting results, the modern methods were used in the work, but the manuscript needs to be improved.

Abstract: (1) Please,  place scientific questions addressed in a broad context and highlight the purpose of the study; (2) Please, place the novelty of the content and highlight the significance of the study.

Introdution should (1) contain scientific questions/hypotheses, (2) related to these questions/hypotheses the purpose of the study.

Conclusions need to be expanded. It should contain the main findings and interpretations, the novelty of the results.

Questions:

Lines 114,122,123, Table 1, Figure 3: KO2 - what is it?

Line 158 : TM- what is it?

Round 2

Reviewer 1 Report

General

The authors addressed most of my comments on the previous version of hte manuscript appropriately, so the manuscript is much improved. There are still numerous writing errors that I highlighted in the Introduction section, so the manuscript needs thorough proofreading once again.

Also, “seedings” is used in many instances in which “seedling” is more appropriate. I used the editing tool of my pdf software to highlight a few of these in the Results section.

Table 1: The authors’ response about “sunshine” is not sufficient for a scientific paper. Sunshine hours is simply not a standard measurement that can be interpreted meaningfully in a scientific paper. The geographic position of the site can provide a fairly accurate estimate of incoming solar radiation. Modification based on cloud cover is certainly important, but sunshine hours cannot be easily translated into anything I am aware of.

Line 120: The authors report N concentration not “total N content”. Also, there is no way N concentration can be 0.0083 g Kg-1. More likely, it is 8.3 g Kg-1, since ~1% N is reasonable for plant tissue concentration.

Figure 1: Inclusion of these histograms is appreciated, but the font size of the titles and axes are too small to easily read, even online.

Figure 2: Font size of axes and mean groupings are still a bit small. Also, best to replace the colors with black. This has greatest contrast and ease of viewing.

Figure 3: Same comments here. In particular, red and green are not recommended because this is the most common form of color-blindness.

Section 3.4: The authors describe correlations that are not significant. In these cases, you cannot confidently claim they are real.

Table 6: I repeat my objection to the inclusion of the correlation of QI with seedling biomass attributes. These biomass attributes are part of the calculation of QI, so their correlation is to be expected.

Page 61, Line 51: I deleted “broad” because this study did not investigate or report on luxury nutrient consumption, sometimes referred to as “nutrient loading”, which intentionally adds nutrients beyond that needed to maximize growth. In this case, the point was to optimize QI based on biomass attributes alone.

Discussion: Sections 4.2-4.4 provide lots of details to things that are stated in brief in Section 4.1 I think the authors should greatly simplify Sections 4.2-4.4, including only the most relevant details needed to support the interpretations in Section 4.1

Also, in the authors’ response to my prior comments, they mention that the T6 treatment has higher biomass even though height is lower because of increased branch (and leaf) biomass. Branch biomass or the effect of fertilization on branch number or growth is not mentioned anywhere in the manuscript. 
